# Evaluation of laser induced sarcomere micro-damage: Role of damage extent and location in cardiomyocytes

**Dominik Müller** [1,2,3]*, **Thorben Klamt**[1,3], **Lara Gentemann**[1,3], **Alexander Heisterkamp**[1,2,3], **Stefan Michael Klaus Kalies**[1,2,3]

**1** Institute of Quantum Optics, Leibniz University Hannover, Hannover, Germany, **2** REBIRTH Research Center for Translational Regenerative Medicine, Hannover, Germany, **3** Lower Saxony Centre for Biomedical Engineering, Implant Research and Development (NIFE), Hannover, Germany

* d.mueller@iqo.uni-hannover.de

**Data Availability Statement:** All relevant data are within the manuscript and its Supporting Information files.

**Funding:** This study was funded by the REBIRTH Research Center for Translational Regenerative

## Abstract

Whereas it is evident that a well aligned and regular sarcomeric structure in cardiomyocytes is vital for heart function, considerably less is known about the contribution of individual elements to the mechanics of the entire cell. For instance, it is unclear whether altered Z-disc elements are the reason or the outcome of related cardiomyopathies. Therefore, it is crucial to gain more insight into this cellular organization. This study utilizes femtosecond laser-based nanosurgery to better understand sarcomeres and their repair upon damage. We investigated the influence of the extent and the location of the Z-disc damage. A single, three, five or ten Z-disc ablations were performed in neonatal rat cardiomyocytes. We employed image-based analysis using a self-written software together with different already published algorithms. We observed that cardiomyocyte survival associated with the damage extent, but not with the cell area or the total number of Z-discs per cell. The cell survival is independent of the damage position and can be compensated. However, the sarcomere alignment/orientation is changing over time after ablation. The contraction time is also independent of the extent of damage for the tested parameters. Additionally, we observed shortening rates between 6–7% of the initial sarcomere length in laser treated cardiomyocytes. This rate is an important indicator for force generation in myocytes. In conclusion, femtosecond laser-based nanosurgery together with image-based sarcomere tracking is a powerful tool to better understand the Z-disc complex and its force propagation function and role in cellular mechanisms.

## Introduction

A high degree of cellular cytoskeletal organization is needed to perform work and generate force in the contractile cells of the heart, the cardiomyocytes (CMs). An impairment of this specific structure-function relationship has been associated with cardiac dysfunction and heart failure [1, 2]. As a popular example, mutation-induced misalignment of sarcomeric integrity was found in the heterogeneous group of cardiomyopathies [3].

Medicine (ZN3440, State of Lower Saxony, Ministry of Science and Culture (Nieders. Vorab)). The funders had no role in study design, data collection and analysis, decision to publish, or preparation of the manuscript.

**Competing interests:** The authors have declared that no competing interests exist.

In 2015, 2.5 million people suffered from cardiomyopathy and myocarditis repercussions [4]. Over the years, the lateral borders (Z-discs) of sarcomeres, the smallest contractile units in CMs, were discovered as critical areas for cardiomyopathy-causing mutations [5]. The multi-protein Z-disc complex is crucial for lateral and longitudinal force transmission [6] and simultaneously acts as a signaling hub to regulate cellular functions [7, 8]. Hence, it has become evident, that knowledge of the fundamental cytoarchitecture and mechanical properties of CMs are prerequisites to explain the causes and consequences of cardiomyopathies [9]. Only a complete understanding of the contractile apparatus and the events occurring at the sarcomere level will improve our knowledge of the whole heart in healthy and diseased conditions.

In former studies, we established a femtosecond (fs) laser-based system to manipulate CMs with sub-micrometer precision, by introducing spatially confined micro-damage to the cells [10–12]. Pulsed laser beams in the fs regime allow a precise, non-invasive ablation of cellular elements without any out off-focus interactions with the tissue [13]. The ablation of structures is achieved by focusing the fs laser beam with a high numerical aperture into the sample to generate a low-density plasma, which results in free-electron-mediated bond breaking. The laser-mediated thermal and mechanical energy transfer is negligible and does not influence the surrounding tissues [13, 14]. In comparison to chemical treatments or overexpression studies, which alter cellular elements in identical quantities [5, 15], this approach allows us to directly analyze cellular repair kinetics. In our recent study, we observed high cell viability and normal calcium homeostasis after ablation of a single Z-disc per CM. Furthermore, the endogenous repair of the initial sarcomeric pattern was detected in more than 40% of treated CMs within hours.

In this study, based on our initial analysis, we utilized the same laser system to firstly analyze the relationship between the damage extent, representing the number of ablated Z-discs, and cell survival. Secondly, we investigated if the location of the damage is associated with cell viability. We analyzed CMs, which received damage central and peripheral to the cell nucleus. In addition, we accounted for multinucleated cells. We evaluated the viable cells after treatment and compared the distribution of cell area and Z-disc number before and after laser treatment. Additionally, we recorded videos of CMs before and after Z-disc ablations to compare sarcomeric organization and CM contractility at different time points. All data underwent a sophisticated image analysis pipeline: We used a self-written software to access the number of Z-discs, SarcTrack to analyze the contractile behavior, and the scanning gradient Fourier transformation to examine the sarcomeric cytoarchitecture.

## Methods

### Neonatal rat cardiomyocyte isolation and culture

Rat cardiomyocytes were isolated from postnatal Spraque-Dawley rats (P2-P5) of both sexes as previously described [11]. One million cardiomyocytes were seeded into 35 mm glass bottom dishes (Ibidi, Germany) for imaging and ablation experiments. Glass bottom dishes were plasma treated (High-Frequency Generator BD-20A, ETP, USA) and coated with 0.1% gelatin 2 h beforehand. After incubation at 37˚C and 5% $CO_2$ atmosphere, cells were washed twice with Dulbecco's phosphate-buffered saline (DPBS, without $Ca^{2+}$, $Mg^{2+}$) the following day and cultured in new medium. CMs were cultured in MEM Eagle medium (PAN Biotech, Germany) supplemented with 5% fetal bovine serum, 100 U/mL penicillin/streptomycin, 0.1 mM bromodeoxyuridine, 1.5 μM vitamin B12, and 1x non-essential amino acids. The medium was exchanged every 2–3 days. CM experiments were performed 8–10 days after isolation. The experiments were in accordance with the German Animal Welfare Legislation (§4, TierSchG) and approved by the local Institutional Animal Care and Research Advisory Committee and

permitted by the Lower Saxony State Office for Consumer Protection and Food Safety (reference number 42500/1H).

## Cardiomyocyte transfection and transduction

In the first set of experiments, rat CMs were transfected with tdTurboRFP-Alpha-Actinin-19 (Addgene plasmid #58050, a gift from Michael Davidson), to visualize the Z-discs in cells. Therefore, 8 μL of ViaFect™ transfection reagent (Promega, Germany) was diluted in 100 μL Opti-MEM (Gibco, Germany) and mixed with 2 μg tdTurboRFP-Alpha-Actinin-19 in 100 μL Opti-MEM. After incubation at room temperature for 20 min, the ViaFect™:DNA solution was added dropwise to the cells (2 μg DNA per dish). The medium was replaced by new medium the following day and experiments were performed two days after transfection. To simultaneously visualize cardiomyocytes' nuclei and Z-discs, cardiomyocytes were transduced with lentiviral particles (pLenti CMV Neo DEST mCherry-H2A-10 and pLenti CMV GFP-ACTN2 Puro). The pLenti CMV GFP-ACTN2 Puro was generated and kindly provided by Christopher S. Chen [16]. The pLenti CMV Neo DEST mCherry-H2A-10 plasmid was generated by cloning the mCherry-H2A-10 construct (Addgene plasmid # 55054, a gift from Michael Davidson) into the destination vector pLenti CMV Neo DEST (Addgene plasmid #17392, a gift from Eric Campeau and Paul Kaufman) via Gateway® Cloning Technology (Invitrogen, USA). Viral particles were produced with a 3rd-generation split packaging system in 293T cells (DSMZ, Germany) using calcium phosphate transfection as previously described [17]. After 48 and 72 hours, the viral supernatant was collected, concentrated by centrifugation at 100 000 g for 2 h at 4˚C and resuspended in DPBS. Lentiviral titers were determined by primers targeting the woodchuck hepatitis virus posttranscriptional regulatory element (WPRE-For: `AGCTATGTG GATACGCTGCTTTA` and WPRE-Rev: `AGAGACAGCAACCAGGATTTATAC`) using the Luna® Universal One-Step RT-qPCR Kit (NEB, Germany). We reproducibly obtained up to 1.05 x $10^8$ infectious units (IU) per mL for pLenti CMV GFP-ACTN2 Puro and 1.07 x $10^8$ IU/mL for pLenti CMV Neo DEST mCherry-H2A-10. Cardiomyocytes were transduced with a multiplicity of infection (MOI) of one overnight and the medium was exchanged the following morning. Z-disc ablation experiments were performed 48 hours post-infection.

## Laser setup, cell imaging and manipulation

A Ti:Sapphire laser system with a pulse length of 140 fs and a repetition rate of 80 MHz was used for multiphoton imaging and Z-disc ablation [11]. The custom-build setup was further equipped with a High-Power LED (SOLIS-3C, Thorlabs, USA) and a CCD camera (ProgRes® MF<sup>cool</sup>, Jenoptik, Germany) to allow epi-illumination and recording of CMs using fluorescence microscopy. This allowed higher imaging speed compared to multiphoton microscopy. Before each experiment, the laser power entering the microscope was adjusted to guarantee equal experimental settings. Fluorescent Z-discs expressing cardiomyocytes were randomly selected with a motorized stage (Prior Scientific, Cambridge, UK) and positions were saved. Multiphoton imaging of tdTurboRFP-Alpha-Actinin-19 or mCherry-H2A-10 was performed at an excitation wavelength of 730 nm. The fluorescence was detected via a photomultiplier tube (Hamamatsu Photonics, Japan) and an emission filter at 607 ± 18 nm. GFP-actinin expressing CMs were imaged with an excitation wavelength of 870 nm and detected with an emission filter at 510–560 nm. Image acquisition with the CCD camera was performed with a dsRed filter cube for tdTurboRFP-Alpha-Actinin-19 and mCherry-H2A-10. GFP-actinin fluorescence was detected with a GFP filter cube. Before nanosurgery, randomly selected CMs were imaged with the CCD camera and via multiphoton microscopy. For nanosurgery, Z-discs were ablated at a wavelength of 730 nm, a scanning velocity of 100 μm/s, and a laser

power of 0.9 nJ as revealed in our recent study. These parameters were validated to result in Z-disc loss via subsequent immunostaining against α-actinin [11]. A minimum of three treated and untreated CMs were recorded per trial. All data were determined from at least three replicates per experiment and condition. All imaging data were analyzed using Fiji [18] and MatLab (MathWorks, Natrick MA, USA, version 2020b).

## Ablation of multiple Z-discs

For multiple Z-discs ablations, positions of tdTurboRFP-Alpha-Actinin-19 expressing CMs were saved and different Z-disc numbers (3, 5, or 10) were selected in a neighboring, longitudinal set or randomly diversified over the whole cell and ablated. Images were recorded before, 10 s, 1 h, and 2 h after ablation. Untreated CMs served as control cells. The viability of CMs was assessed 2 h post nanosurgery by adding 2 μM Calcein-AM to the culture medium. In preliminary experiments, we identified this period as sufficient to identify dead cells. To prove this hypothesis, we performed the same experiment as described above and ablated three Z-discs per CM but analyzed the metabolic activity after 24 h. Recorded images were analyzed equally and compared with viability data 2 h post nanosurgery.

After an incubation time of 20 min at 37°C and 5% $CO_2$ atmosphere, the Calcein fluorescence, which indicates an active cell metabolism, was determined and images were taken with the CCD camera. Non-fluorescent cells were counted as dead cells. Recorded images were further processed to identify if the number of Z-discs is relevant to the damage response. For this purpose, we developed an OpenCV based Python application to quantify the number of Z-discs within the cell area before nanosurgery. In short, Fourier transformation, automated thresholding, and edge-detection based methods were implemented to detect and count the Z-discs (S1 Fig). Furthermore, the visualized cytoskeleton of CMs was encircled with Fiji to calculate the ratio of cell area to Z-discs per cell.

## Central or distal Z-disc ablation

In a randomly selected portion of mCherry-H2A-10 and GFP-actinin expressing CMs, a single Z-disc per cell (proximal or distal to the nucleus) was ablated. Untreated CMs served as control cells. Images of the Z-disc pattern were recorded before and every hour after nanosurgery. The distance between the ablated Z-disc and the center of the nearest nucleus was measured using Fiji. The metabolic activity was assessed after 24 h. Non-fluorescent cells were counted as dead cells.

## Cell analysis using SarcTrack and Scanning Gradient Fourier Transformation (SGFT)

To identify the impact of multiple Z-discs ablations on the sarcomere shortening, the contraction and relaxation time, and the contraction period, we used a MatLab based software algorithm (SarcTrack) recently published by Toepfer *et al.* [19]. Video sequences of GFP-actinin expressing CMs were recorded before, 1 min, and 2 h after ablation of different Z-disc numbers (1, 3, or 5) at a frame rate of 4 FPS. Videos were optimized by bleach correction, background subtraction, and enhanced local contrast adjustment using Fiji. The SarcTrack parameters were optimized such that the paired wavelet matching method reliably identified sarcomeres in 92% of all CMs.

Furthermore, images of contracted and relaxed CMs were extracted from the recorded videos to determine sarcomere orientation and alignment before and after Z-disc removal. For this analysis, a MatLab package for the scanning gradient Fourier transform (SGFT) method recently published by Salick *et al.* was utilized [20].

### Data analysis and statistics

All data sets were analyzed and graphically represented using Origin (OriginLab, USA, version OriginPro 2018b) or MatLab. Viability data were expressed as mean + standard deviation (SD) and analyzed with a One-Way ANOVA followed by post-hoc Tukey t-test analysis. The data obtained for the cell areas and Z-disc counts were plotted as Violin plots, excluding outliers. A Kolmogorov-Smirnov test was used to detect statistically significant distributions. The Sarc-Track data were depicted as raw data and analyzed with a Two-Way ANOVA considering the point in time and the number of ablated Z-discs. Post-hoc analysis was performed using a Holm-Bonferroni test. The sarcomeric angular distributions, obtained from the SGFT measurements, were either depicted as an angular distribution (single cell at different points in time) or as raw data (all measurements). To test if the angular distribution changed significantly post or 2 h after ablation in relation to the pre-ablation state, a Watson-Williams-test (MatLab circular statistics toolbox [21]) was performed comparing these three datasets for every single cell. Additionally, we aimed to analyze the absolute alignment change for all cells with a Two-Way ANOVA considering the point in time and the number of ablated Z-discs. Post-hoc analysis was performed using a Holm-Bonferroni test.

For all experiments, all cells per dish were randomly selected and at least three independent dishes per experimental condition were used. In all cases, p-values < 0.05 were considered as statistically significant.

## Results

### Cardiomyocyte viability associates with the extent of Z-disc damage within 2 h after damage

To determine if the extent of Z-disc removal relates with CMs viability, we ablated multiple Z-discs per CM. We compared the metabolic activity with Calcein-AM 2 h after ablation of 3, 5, or 10 Z-discs per cell. Furthermore, the Z-discs were either ablated in a longitudinal set (Fig 1A) or randomly diversified over the cell´s Z-disc pattern (Fig 1B). We detected a significant reduction of CM viability after ablation of 10 Z-discs to 16% for neighboring Z-discs (Fig 1C, p ≤ 0.006) and 9% for randomly selected Z-discs (Fig 1D, p ≤ 0.003) compared to untreated control CMs. Besides, we observed that the ablation of 10 Z-discs leads to cell death within minutes after nanosurgery, independent of the damage pattern (S2 Fig). Ablation of 5 neighboring Z-discs led to a CMs survival of 50%, while a significant viability decrease (p ≤ 0.02) to 31% was detected for randomly ablated Z-discs. A non-significantly decreased viability was observed after ablation of 3 Z-discs.

### Cell area and total Z-disc count are not critical for CM survival

Due to the strong interconnection of Z-discs in CMs, we hypothesized that smaller CMs are more susceptible to Z-disc ablation, compared to large CMs. To examine this hypothesis, the cell area of all CMs was analyzed before and after the ablation of multiple Z-discs (Fig 2). We did not observe a significant relationship between cell area and CM survival, neither for ablation of Z-discs in a set (p ≥ 0.14) nor for randomly selected Z-discs (p ≥ 0.81) within the cell. We also counted the Z-discs within the cell area and could not detect a significant influence of the Z-disc count (p ≥ 0.09) on the cell survival.

### Multiple Z-disc damage induces immediate CM death

In the next series of experiments, we analyzed the time-dependent survival rate of CMs post nanosurgery. 3 Z-discs per CM were either ablated in a longitudinal set or randomly selected

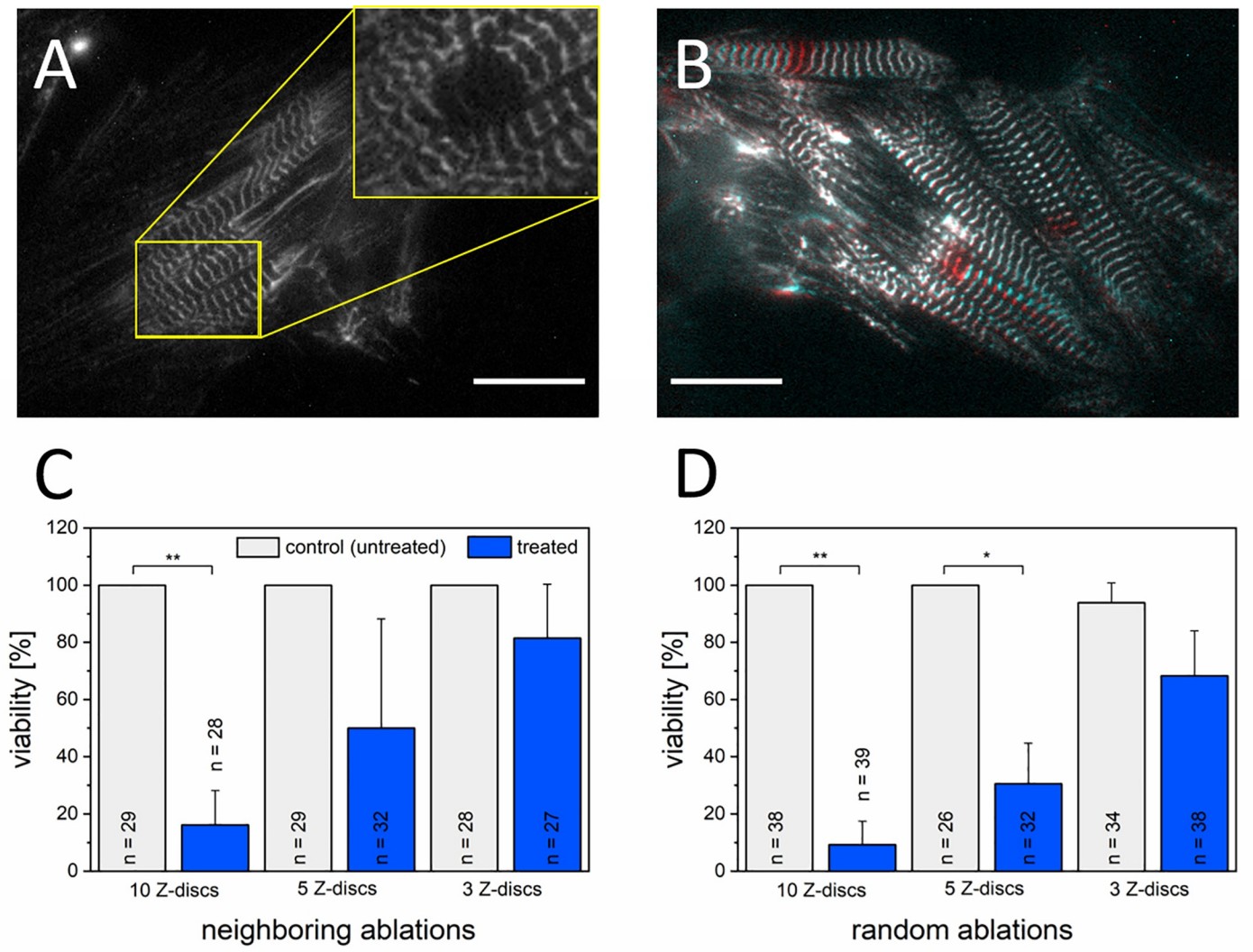

**Fig 1. Relationship between CMs viability and the extent of Z-disc damage within 2 h after damage.** Femtosecond laser-based nanosurgery was used to ablate 10, 5, or 3 Z-discs per CM. Neighboring Z-discs were ablated in a longitudinal set (A, C) or randomly selected Z-discs were ablated (B, D) over the cell. A representative turboRFP linked α-actinin expressing rat CM before and directly after the ablation of 3 Z-discs in a set (yellow box) is depicted in (A). A merged image before (red) and after (cyan) random ablation of 3 Z-discs is shown in (B). Scale bars 20 μm. Bar charts represent mean viabilities + standard deviation. Treated CMs were statistically compared with untreated, control CMs (individual determined for each experiment) using a One-Way ANOVA followed by Tukey test ($^{**}p < 0.01$, and $^{*}p < 0.05$).

over the cell´s Z-disc pattern as described before. The metabolic activity was visualized 2 h or 24 h after Z-disc ablation with Calcein-AM (Fig 3). No statistically significant differences in viability were observed for the ablation of randomly selected Z-discs (p = 0.99) as well as for the ablation of neighboring Z-discs (p = 0.59).

In addition, the cell area and Z-disc number of CMs were analyzed before and after ablation of 3 Z-discs for both points in time (Fig 3). Both, the distribution of cell area and Z-disc count were comparable between pre and post ablation state for both time intervals.

## Cardiomyocyte survival is independent of the damage position

To analyze potential differences between peripheral and central micro-damage in CMs, we selectively ablated a single Z-disc per cell at different distances to the nucleus. Cells were double transduced to visualize the nucleus and the Z-disc pattern. Following ablation of a single

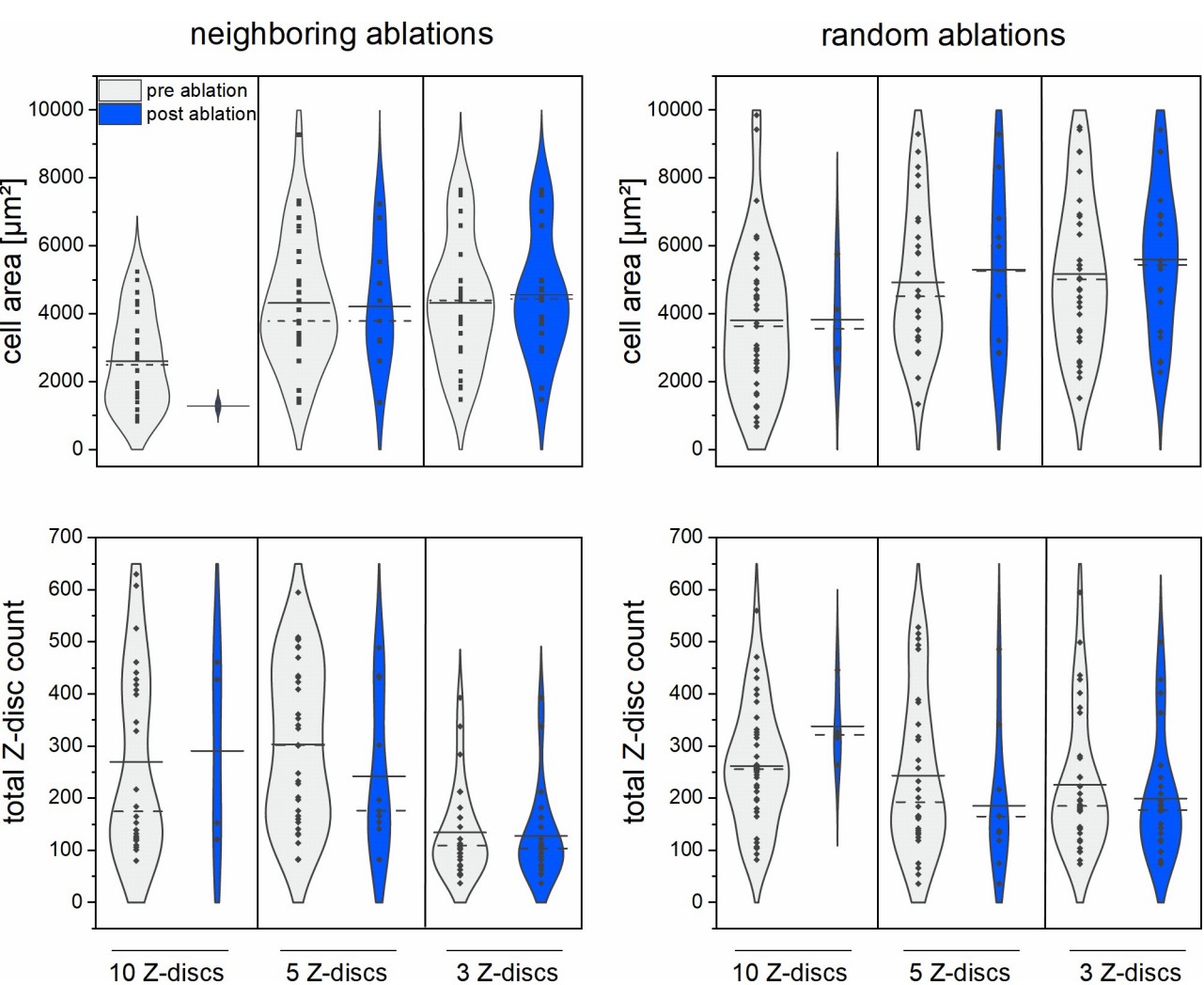

**Fig 2. Relationship between CMs cell areas and total Z-disc counts with cell survival.** Violin plots are displayed to visualize the distribution, median (dotted line) and mean (solid line) of CMs in terms of cell area (upper panel) and total Z-disc counts (lower panel) pre (grey) and 2 h post (blue) multiple Z-discs ablations. Each dot represents a single cell. Neighboring Z-discs in a longitudinal set: 10 Z-discs n = 26, 5 Z-discs n = 32, 3 Z-discs 27; Randomly selected Z-discs: 10 Z-discs n = 39, 5 Z-discs n = 29, 3 Z-discs n = 33.

Z-disc per CM either close to the nucleus or peripheral, the cell metabolism was visualized 24 h after nanosurgery. We observed a viability of treated CMs of 62% (Fig 4A) compared to untreated control CMs (89%). Adjacent cells were unaffected by laser-induced manipulation. Furthermore, the micro-damage distance to the nucleus was not significantly influencing the viability of CMs (p = 0.58, Fig 4B).

We also assessed whether the number of nuclei and the damage distance to the nucleus are related to cell survival (Fig 4C, 4D). In CMs with two nuclei, the distance to the ablated Z-disc was determined to the nearest nucleus. No statistically different viabilities were observed (p ≥ 0.48).

## Sarcomere reorientation during contraction

A critical factor for force generation in cardiomyocytes is a homogeneous sarcomere alignment. To analyze the impact of micro-damage to the myofibril integrity, we quantified the

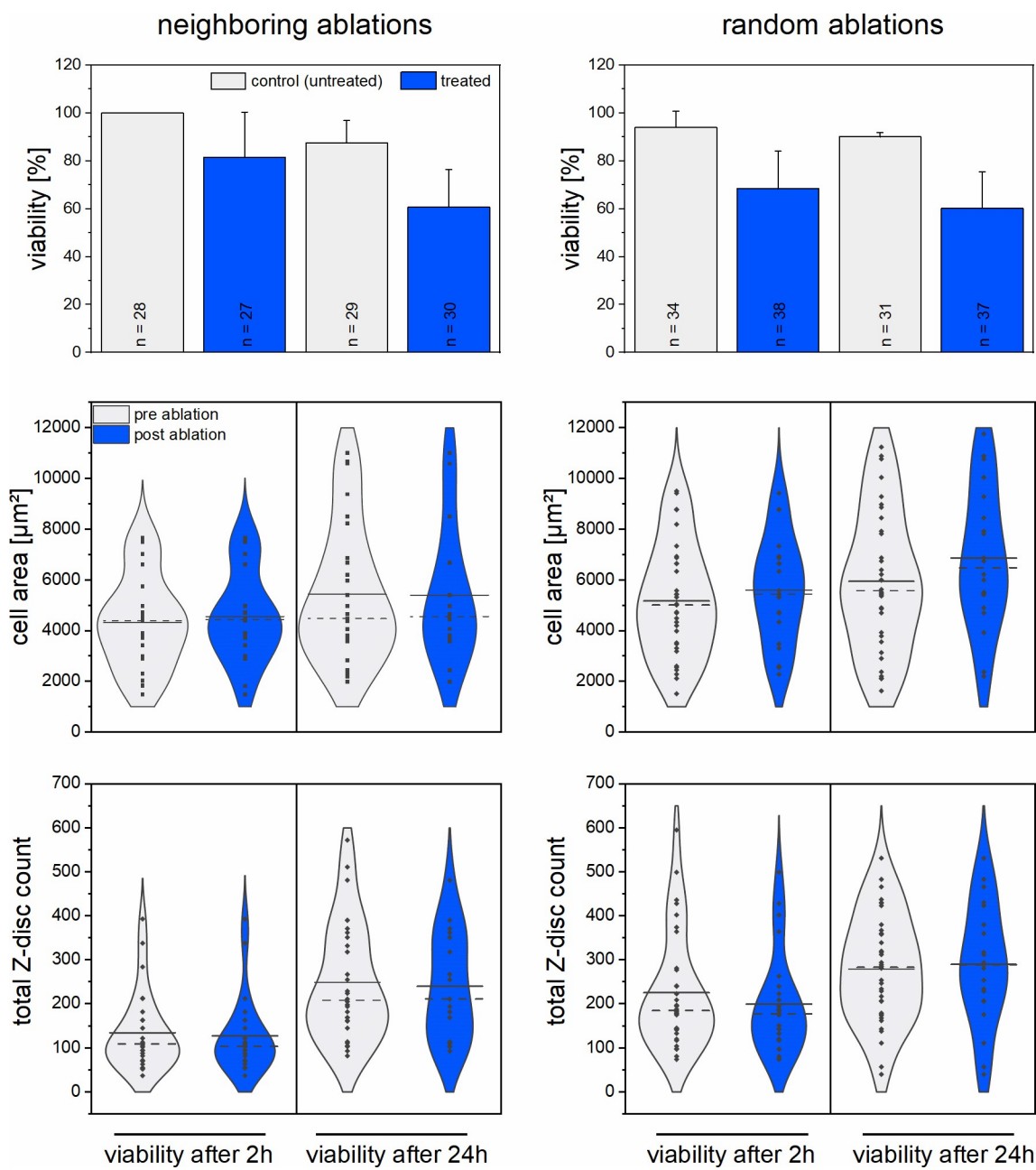

**Fig 3. Comparison of CM survival after 2 h and 24 h post ablation of 3 Z-discs.** Bar charts represent the mean viabilities + standard deviation of CMs. In addition, either 3 neighboring Z-discs or 3 Z-discs randomly selected over the cell were ablated. Violin plots contrasting the distribution, median (dotted line), and mean (solid line) of CMs in terms of cell area (middle panel) and total Z-disc counts (lower panel) pre (grey) and post (blue) multiple Z-disc ablations. Each dot represents a single cell. Neighboring Z-discs in a longitudinal set: viability after 2 h area n = 27, Z-disc count n = 27, viability after 24 h area n = 29, Z-disc count n = 30; Randomly selected Z-discs: viability after 2 h area n = 33, Z-disc count n = 33, viability after 24 h area n = 34, Z-disc count n = 36.

level of sarcomere organization by recording videos pre, 1 min, and 2 h post Z-disc ablation. Neighboring Z-discs were ablated and either 1, 3, or 5 ablations were performed. We calculated the direction and orientation of the myofibril network using a scanning gradient Fourier transform (SGFT) method [20]. Thereby, local sarcomere orientations were determined and overall alignment was assessed by weighing the direction values. A representative analysis of a

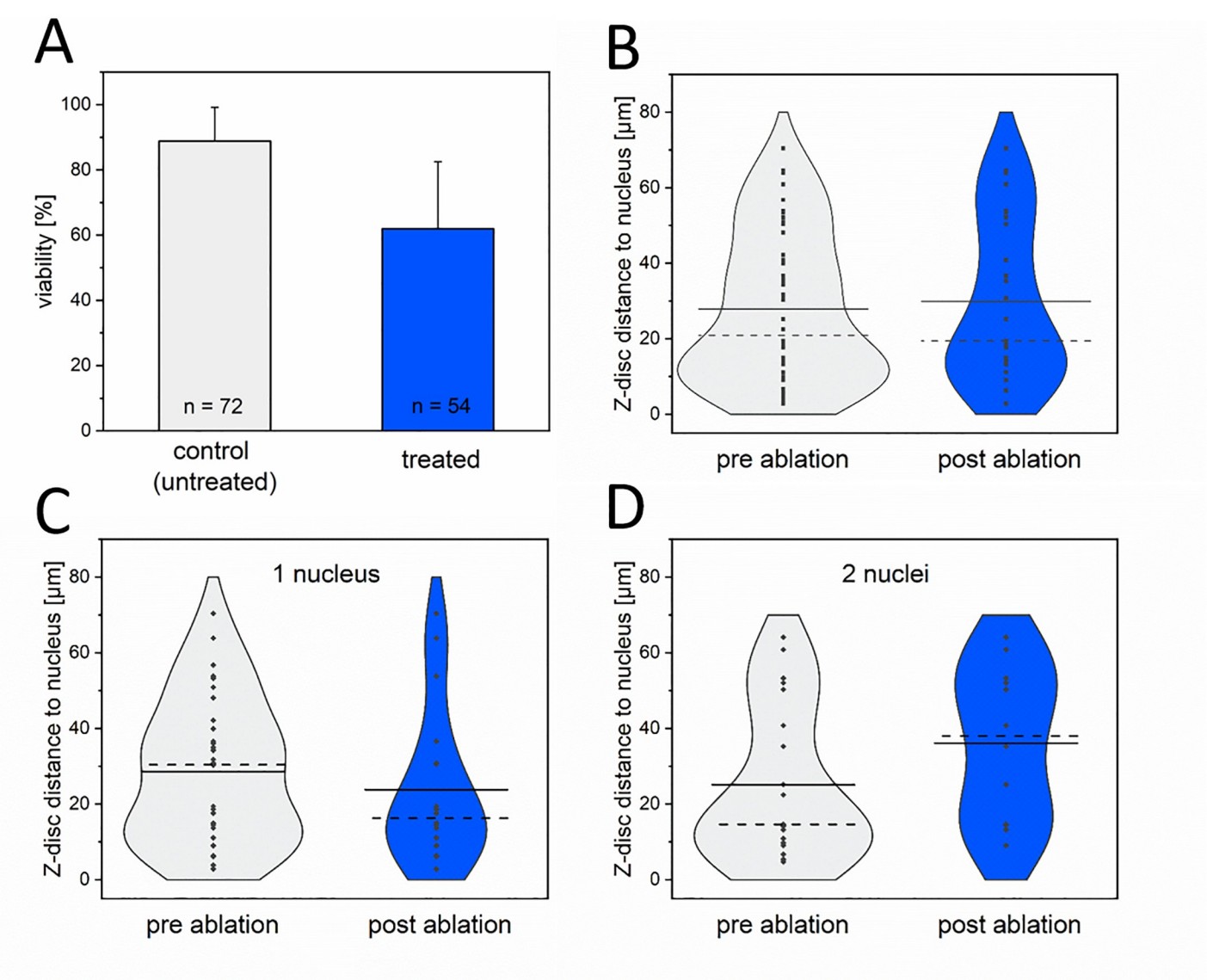

**Fig 4. Relationship between damage position and CM survival.** The viability of CMs 24 h after single Z-disc ablation (A) was determined using Calcein-AM staining. Bar chart represents mean viabilities + standard deviation. Violin plots contrasting the distribution, median (dotted line), and mean (solid line) Z-disc distance to the nucleus of CMs pre (grey) and post (blue) ablation of a single Z-disc per CM. Each dot represents a single cell. 1 nucleus n = 32, 2 nuclei n = 21, (3 nuclei n = 1 not shown). No statistical differences were observed between the pre and post ablation groups for each setting (p ≥ 0.48).

sarcomeric pattern is depicted in Fig 5. The angular distribution was computed from the Z-disc pattern. In addition, the gradient mapping of Z-discs generated a quiver plot of sarcomere directions, and a pattern strength heat map was created to identify regions with repeating patterns. In this example, the overlay of the Z-disc pattern pre (red) and 1 min post-ablation (cyan) clearly indicates the ablation of 5 Z-discs (Fig 5B, yellow arrow). The SGFT analysis did not reveal a significant change in the myofibrillar alignment and orientation between pre and 1 min post-ablation. In contrast, by comparing the pre and the 2 h post nanosurgery patterns, a significant alteration was detected (p = 0.006).

We also compared the angular distributions of myofibril alignment in the relaxed (Fig 6A) and contracted (Fig 6B) cell state of CMs. Therefore, the percentage of sarcomeres within a range of 20°, as a robust angle for disarray detection, was compared to the post ablation

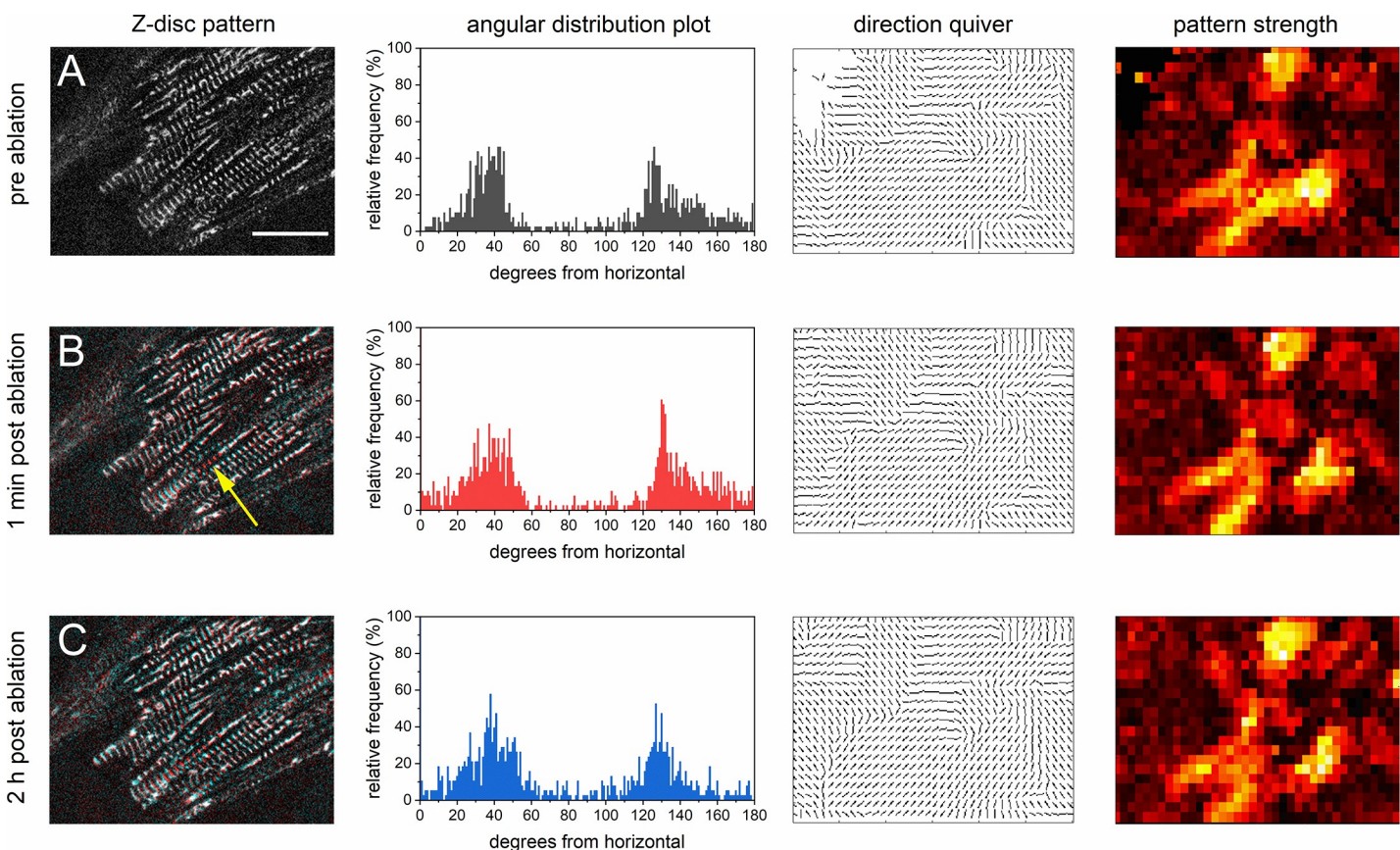

**Fig 5. Sarcomere orientation was analyzed pre and post Z-disc ablation using Scanning Gradient Fourier Transform (SGFT) method.** From a representative Z-disc pattern of a GFP-actinin expressing CM before nanosurgery (A), the angular orientation distribution, the sarcomere direction (quiver plot), and the sarcomere pattern strength were examined. The pattern strength heat map visualized regions with high levels of repeating patterning in yellow/white and less repeating structures in darker red. Following, 5 neighboring Z-discs were ablated (yellow arrow). An overlay of the Z-disc pattern pre (red) and post-ablation (cyan) is shown in (B). The SGFT analysis revealed a non-significant change in the angular distribution (p = 0.45). In contrast, a significant change was found 2 h post-ablation (p = 0.006). Data were statistically compared using a Watson-Williams-test for circular distributions. Scale bar 20 μm.

alignment. The data of all groups were not significantly different, neither for all time points nor for the number of ablated Z-discs. However, the statistical comparison between the angular distributions on a single cell level revealed, that even a single Z-disc ablation could lead to a significantly different angular distribution (S1 Video). This was observed for more than 40% of CMs in the relaxed (Fig 6C) and 67% in the contracted cell state (Fig 6D). This percentage further increased at later time points but not necessarily with the number of ablated Z-discs. However, it needs to be considered that the Watson-Williams-test mainly addresses the means of the distribution.

To elucidate the influence of micro-damage on the contractility and sarcomere shortening in CMs, we applied the software package SarcTrack, which was recently developed by Toepfer *et al.* [19]. Videos of CMs pre and post ablation of Z-discs were processed to track sarcomere displacement, contraction period, relaxation duration, and contraction time. The data demonstrate, that no statistical differences were observed by comparing mean values for contraction time (Fig 7A), contraction period (Fig 7B), and relaxation duration (Fig 7C). The same holds true for the percentage of sarcomere shortening after ablation of 3 and 5 Z-discs. However, the sarcomere shortening decreased significantly (p = 0.035, with a statistical power of 68%) from 7.7% (post ablation) to 4.7% (2 h post ablation) by ablation of a single Z-disc in CMs (Fig 7D).

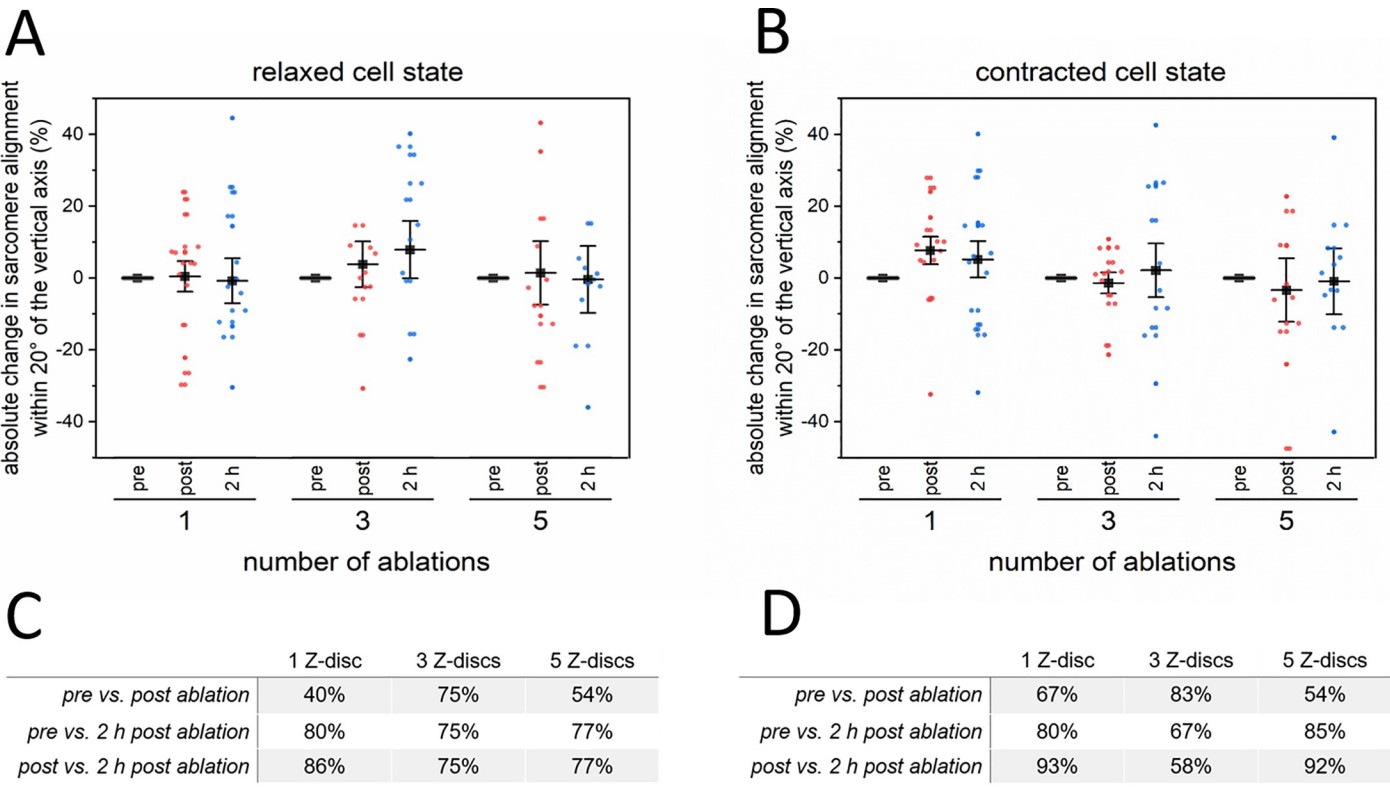

**Fig 6. Quantitative analysis of sarcomere orientation pre and post Z-disc ablation using Scanning Gradient Fourier Transform (SGFT) method.** The sarcomere pattern of GFP-actinin expressing CMs pre, 1 min, and 2 h post nanosurgery were analyzed. The resulting absolute changes in sarcomere orientations within 20° angle of the vertical axis are depicted for the relaxed (A) and for the contracted cell state (B). Each dot represents a single cell. Charts represent mean values ± SEM. The percentage of CMs, which differed significantly (p < 0.05) in their angular sarcomeric orientation on a single cell level, are listed for the relaxed (C) and for the contracted cell state (D). Data were statistically compared using a Watson-Williams-test for circular distributions. 1 Z-disc n = 15, 3 Z-discs n = 12, 5 Z-discs n = 13.

## Discussion

In the field of cardiomyopathies, many studies revealed a pivotal role of sarcomere organization and altered sarcomere structures as key factors during disease development and progression. However, the exact role of isolated sarcomere elements in the orchestra of the CM cytoskeleton is still very patchy. In particular, the role of the Z-disc complex during sarcomere repair has to be investigated. Therefore, this study was designed to evaluate the consequences of physical Z-disc removal in a systematic way. To do this, we used an fs laser-based setup to precisely ablate Z-disc elements in CMs. This setup allowed us to introduce different damage patterns in CMs in a spatio-temporal confined and reliable fashion [11, 12].

In contrast to electric stimulation [22] or mechanical strain [23], which results in irregular sarcomere disruption, we precisely introduced different extent of damage by ablating 3, 5, or 10 Z-discs per CM. The metabolic activity of cells was assessed 2 h post nanosurgery. As expected, we observed a significant dependency between damage extent and CM survival. Increased cytoskeletal damage results in increased mortality of CMs. However, the damage response of CMs is multiphasic and cell death mechanism might depend on the damage extent. In our earlier study, we detected membrane blebbing in some of the treated CMs after ablation of a single Z-disc, which might indicate an apoptosis-like pathway [24, 25]. In the present study, we observed an immediate cell death within minutes after ablation of 10 Z-discs without membrane blebbing. Therefore, it might be that damage over a certain threshold lead to disruption of the sarcolemma, as Z-discs are physically coupled to the sarcolemma via

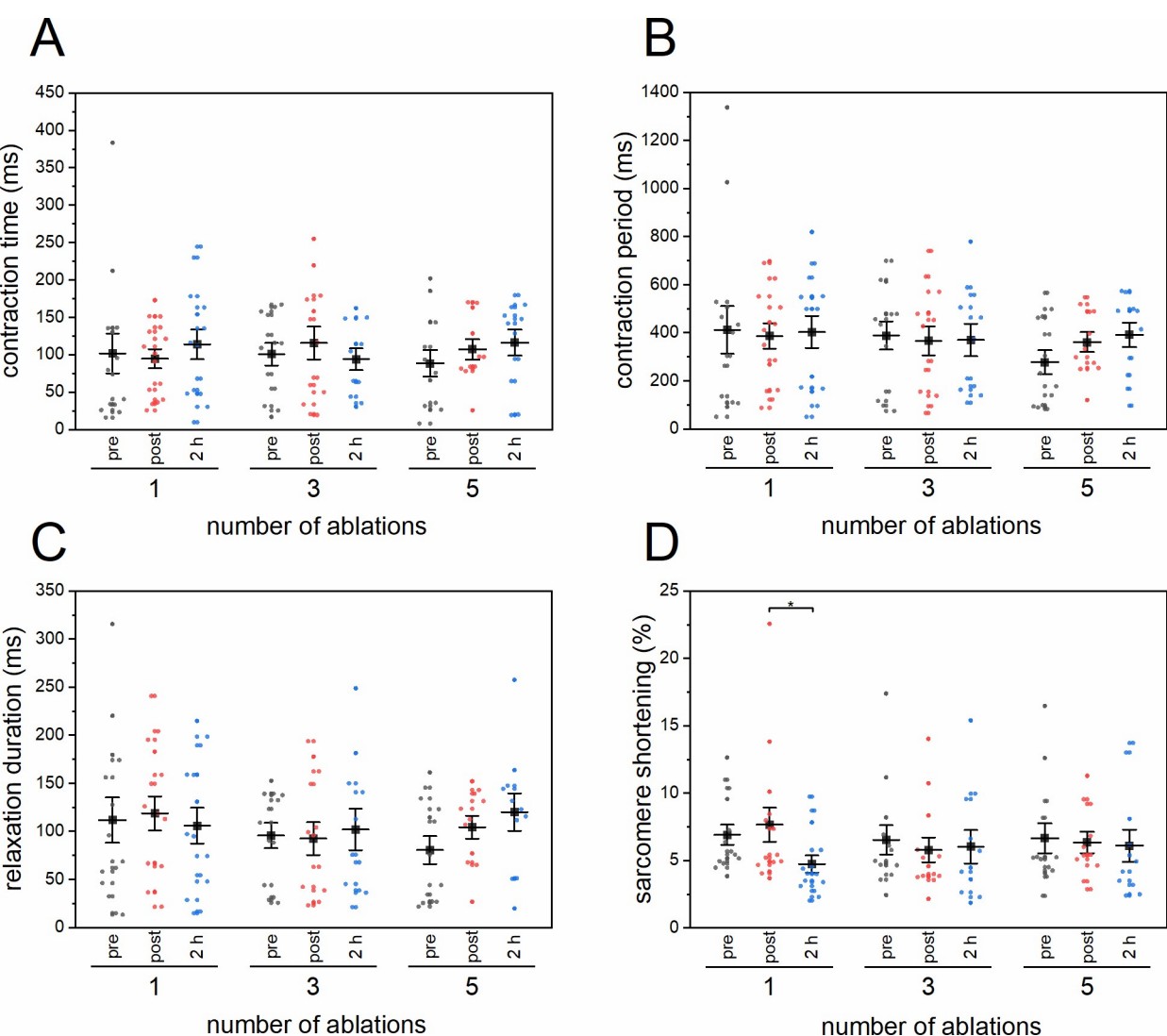

**Fig 7. SarcTrack assessment of CMs pre and post Z-disc ablation.** The sarcomere pattern of GFP-actinin expressing CMs pre, 1 min, and 2 h post nanosurgery were analyzed. CM contractility was compared pre and post ablation of 1, 3, or 5 Z-discs in a longitudinal set in terms of contraction time (A) and contraction period (B). In addition, the relaxation duration (C, time from max contraction to max relaxation) and the sarcomere shortening as a percentage of the initial sarcomere length (D) were plotted. Each dot represents a single cell. Charts represent mean values ± SEM. Data were tested for significance with a Two-Way-ANOVA and Holm-Bonferroni test. *p < 0.05. 1 Z-disc n = 14, 3 Z-discs n = 12, 5 Z-discs n = 11.

costameres [26]. Studies have shown that the sarcolemma of contracting cells is very susceptible to damage [24, 27]. Thus, sufficient damage could result in sarcolemma instability or collapse and leaking of sarcoplasm. If so, this might be characteristic for a necrotic cell death pathway [24]. Nevertheless, this hypothesis needs to be evaluated in further studies. An induced cell death via reactive oxygen species (ROS) is unlikely, as we could not detect significantly elevated ROS formation after laser-induced Z-disc ablations (S3 Fig). Nonetheless, it is possible that the application of the nanosurgery itself will induce further downstream effects, which influence cell survival, in particular, for many Z-disc ablations. However, this cannot be reliably elucidated due to the strong dependence of fs laser-based nanosurgery on the cell type, state and the ablated structure [12].

As critical factors for CM survival after Z-disc removal, we hypothesized that the total number of Z-discs and/or the cell area are essential. Therefore, we measured the cell area and counted the total number of Z-discs. We did not observe a significant relationship between cell area or Z-disc count and CM survival.

The Z-disc complex is, in addition to its force propagation function, a signaling hub for a variety of cellular pathways. For instance, Z-disc proteins like FATZ, ENH or titin are involved in sensing mechanical stress, regulating contraction and downstream phosphorylation of proteins [1]. Furthermore, associated Z-disc proteins can shuttle between the Z-disc complex and the nucleus in CMs. Nuclear translocation was reported for muscle LIM proteins (MLPs) in response to extrinsic pressure overload leading to MLP mediated cell-fate determination and tissue-specific gene expression [28–31]. Hence, we were interested if the distance between Z-disc ablation and nucleus is crucial for CM survival. By comparing the viability pre and 24 h post ablation of a single Z-disc per CM, the median distributions were similar. In earlier studies on glioma cells, regional variations after stress fiber (SF) ablations were observed. Disruption of peripheral SFs can trigger retraction of the whole-cell whereas central SF disrupts consequences only in minor changes [32]. In future studies, our femtosecond laser setup could also be used to clarify the hypothesis, that the addition of new sarcomeres during repair processes is possible throughout the entire length of CMs [23].

As myofibrillar rearrangement occurs during healthy, developing myocardium [33] but disarray after eccentric exercise [34] or in pathogenic phenotypes [35–37], we used the SGFT software to determine sarcomere arrangement. By analyzing the sarcomere orientation pre and post ablation of different numbers of Z-discs, the myofibril alignment in a 20° angle along a single axis was similar to the pre surgery state by comparing all groups. Nevertheless, on a single cell level, we observed significantly different angular distributions by more than 40% of CMs, with an increased percentage at later time points but not necessarily with the number of ablated Z-discs. Studies have shown that damaged sarcomere regions can enlarge from one or more sarcomeres, which will lead to disorientation of the original pattern [22, 23]. However, the main focus in this study is on the single cell level. Therefore, it is impossible to relate our findings to specific disease conditions.

As homogenous and controlled contraction is essential for sufficient force production but altered pathophysiological during diseases [38–40], we analyzed sarcomere dynamics of CMs with SarcTrack. We revealed a constant contraction time, contraction period, and relaxation duration after nanosurgery. These findings are in agreement with earlier studies by Leber *et al.* [41] and with calcium measurements after ablation of a single Z-disc in CMs [11]. Continuous contraction has also been reported as a critical factor for proper myofibril assembly [15], which is required during sarcomere repair. In addition, these observations suggest a stabilization of the damaged region, as found for Filamin C [41] or F-actin [23], without relinquishment of contractility. Nevertheless, earlier studies indicate that damaged regions are stretchable but non-contractile, resulting potentially in reduced tension or force transmission in CMs [22]. An important indicator for force generation in myocytes is the change in sarcomere length (SL) during contraction [42]. We observed shortening rates between 6–7% of the initial sarcomere length. While maximal force is generated close to resting sarcomere length [43], it might be that force generation is not significantly affected after Z-disc ablations. Nevertheless, further direct methods [44–46] should be performed to quantify local force generation after laser-based manipulation on a single cell level. The significant reduction of sarcomere shortening in case of ablation of a single Z-disc might speculatively be related to faster endogenous repair. However, this needs to be analyzed in future experiments.

The architecture, composition, and functionality of cardiomyocyte cytoskeleton still lacks data on the mechanisms involved in the repair and reassembling of these complex structures.

In particular, the exact nature of sarcomere damage response for defined damage patterns and regions remains under-characterized. In this study, we showed that CM survival associates with the damage extent, but not with the the cell area or total number of Z-discs. Furthermore, CM survival is independent of the damage position and can be compensated. This might be due to sarcomere reorganization during ongoing contraction [41]. In conclusion, femtosecond laser-based nanosurgery in combination with software-based sarcomere tracking is a powerful tool to investigate CM repair scenarios and improvement of these on a single cell level. This will accelerate cardiovascular research from mechanistic discoveries to therapeutic applications.

## Supporting information

**S1 Fig. Automated Z-disc count in cardiomyocytes.** An OpenCV application was developed to quantify the total Z-disc count by single CMs. Therefore, unprocessed images were Fourier transformed, thresholded and via edge-detection refined to allow Z-disc counts in the processed layer. The representative image depicts the different processing layers and an example of a selected area and the resulting count of 5 Z-discs with a next-neighbor-distance of 1.83 μm.
(TIF)

**S2 Fig. Rapid cell death of CMs after 10 Z-discs ablations.** Using Calcein-AM dye, this representative image series depicts an active cell metabolism of a CM before (A) and a rapid cell death within minutes (D-F) after nanosurgery. A turboRFPlinked α-actinin expressing CM is shown before (A) and after ablating of 10 neighboring Z-discs (C, damage region indicated by the yellow arrow). 15 minutes post ablation (F), the treated CM did not show an active cell metabolism (white arrow). Accumulation of Calcein in untreated cells was found in all experimental trials after an extended incubation time with Calcein-AM. Scale bar 25 μm.
(TIF)

**S3 Fig. Negligible generation of reactive oxygen species after Z-disc ablations.** CMs were transduced with pLenti CMV mApple-ACTN2 Puro to visualize the Z-disc pattern and loaded with 10 μM 5-(and-6)-chloromethyl-2′,7′-dichlorodihydrofluorescein diacetate, acetyl ester (CM-H$_2$DCFDA; Invitrogen, USA) for 30 min in DPBS with Ca$^{2+}$ and Mg$^{2+}$ to detect reactive oxygen species (ROS) after nanosurgery. After washing twice with Opti-MEM (Gibco 11058–021, Germany), Z-disc pattern and the basal fluorescence of H$_2$DCFDA, at a wavelength of 850 nm, were recorded. Following ablation of different numbers of Z-discs, H$_2$DCFDA fluorescence was assed directly, 1 min and 2 min post ablation. A representative time series for the ablation of one and five Z-discs (marked by arrows) and the region were H$_2$DCFDA fluorescence was assed (dotted circle, pseudocolor) is shown in (B). H$_2$DCFDA fluorescence was normalized to the pre ablation level and non-significant ROS generation was found after ablation of Z-discs in all treated cells (C) as well as in neighboring cells (D). A significant ROS increase was observed in the positive control, where 100 μM H$_2$O$_2$ was added (A). Untreated CMs served as negative control. Scale bars 10 μm. Bar charts represent relative fluorescence intensities (%) + standard deviation.***p $<$ 0.01, One-Way ANOVA followed by Tukey test. ctr. n = 14, 1 Z-disc n = 20, 3 Z-discs n = 19, 5 Z-discs n = 14, 10 Z-discs n = 17.
(TIF)

**S1 Video. Contracting cardiomyocyte before and after ablation of a single Z-disc.**
(AVI)

## Acknowledgments

We thank Jasmin Bohlmann and Jennifer Harre from NIFE for providing the neonatal rat puppies. We also acknowledge Prof. Christopher S. Chen and his group from the Department of Biomedical Engineering, Boston University, USA, for kindly providing the pLenti CMV GFP-ACTN2 Puro and the pLenti CMV mApple-ACTN2 vectors.

## Author Contributions

**Conceptualization:** Stefan Michael Klaus Kalies.

**Data curation:** Dominik Müller, Thorben Klamt, Stefan Michael Klaus Kalies.

**Formal analysis:** Dominik Müller.

**Funding acquisition:** Alexander Heisterkamp, Stefan Michael Klaus Kalies.

**Investigation:** Dominik Müller, Thorben Klamt, Lara Gentemann.

**Methodology:** Stefan Michael Klaus Kalies.

**Project administration:** Alexander Heisterkamp.

**Software:** Thorben Klamt.

**Supervision:** Alexander Heisterkamp, Stefan Michael Klaus Kalies.

**Validation:** Dominik Müller, Lara Gentemann, Stefan Michael Klaus Kalies.

**Visualization:** Dominik Müller, Stefan Michael Klaus Kalies.

**Writing – original draft:** Dominik Müller, Stefan Michael Klaus Kalies.

**Writing – review & editing:** Dominik Müller.

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
