## [Decision Letter · Decision Letter 0]

9 Feb 2021

PONE-D-21-00038

Evaluation of laser induced sarcomere micro-damage: role of damage extent and location in cardiomyocytes

PLOS ONE

Dear Dr. Müller,

Thank you for submitting your manuscript to PLOS ONE. After careful consideration, we feel that it has merit but does not fully meet PLOS ONE’s publication criteria as it currently stands. Therefore, we invite you to submit a revised version of the manuscript that addresses the points raised during the review process.

We look forward to receiving your revised manuscript.

Kind regards,

Xiaolei Xu

Academic Editor

PLOS ONE

Journal Requirements:

Reviewers' comments:

Reviewer's Responses to Questions

**Comments to the Author**

1. Is the manuscript technically sound, and do the data support the conclusions?

Reviewer #1: Partly

Reviewer #2: Yes

2. Has the statistical analysis been performed appropriately and rigorously? 

Reviewer #1: I Don't Know

Reviewer #2: Yes

3. Have the authors made all data underlying the findings in their manuscript fully available?

Reviewer #1: Yes

Reviewer #2: Yes

4. Is the manuscript presented in an intelligible fashion and written in standard English?

Reviewer #1: Yes

Reviewer #2: Yes

5. Review Comments to the Author

Reviewer #1: The manuscript PONE-D-21-00038 “Evaluation of laser induced sarcomere micro-damage: role of damage extent and location in cardiomyocytes” by Müller et al evaluates laser-induced Z-disc damage. Previously (Sci Rep 2019), the group reported that ablation of a single Z-disc in a single cell did not affect viability or calcium homeostasis. In the current study, the authors compare ablation of a single, three, five, or ten Z-disc structures in a single cell. They report that ten ablations cause cell death and multiple ablations cause myofibril disarray and alterations in sarcomere shortening.

Major/Interpretation Concerns

1) The authors note that 10 cell ablations do cause cellular death, but do not know the mechanism nor indicate that total power in the cell was controlled.

A) This reviewer is concerned that the mechanism itself may be important to the interpretation of the author’s data. In both the earlier study and the current study, the authors do not study reactive oxygen species generation (although they clearly note it as a possible consequence in the 2019 study). The authors also do not shot show conclusively that Z-disc proteins are ablated, only that fluorescence is eliminated then re-formed. Thus, this reviewer is concerned that the authors are merely de-activating the fluorescence by causing the RFP or mCherry to ablate. Could the ablation itself and/or the laser input be causing ROS? If this is true, then the authors are studying local ROS generation instead of Z-disc dysfunction. Such a mechanism may also provide a mechanism for cell death in the 10 ablation strategy. This reviewer suggests measuring the ROS production in the cells.

If the authors cannot generate such data, this reviewer suggests that the authors provide a detailed comparison between their findings and findings in pathophysiological states that may alter Z-disc alignment, such as muscular dystrophy or nemaline myopathies.

B) While the authors note that the laser power settings were equalized for each experiment (Methods lines 151/152), they do not indicate that total power/energy delivery into the cell was normalized. This reviewer suggests: controlling for absolute power/energy delivered into the cell (for example 100 small bursts of 10% of the ablation pulse through the cell) to determine if cell death is based the total power/energy delivered to the cell.

Note that the reviewer believes that this may also provide evidence for or against the hypothesis that ROS production causes the cell death and damage

2) Discussion Paragraph around Line 422. The authors suggest that Refs 22, 29-31 indicate that myofibril disarray occurs. To this reviewer’s reading, they refer primarily to cellular alignment (for example, Ref 31 discusses cellular myoFIBER alignment, not myoFIBRIL alignment. Reference 22 is an invitro model, not a true disease model. Only a very small statement in reference 29 about the pre-myofibril is clearly indicative of alignment.

The authors should clearly note this limitation or provide stronger justification that such a small number of Z-discs would ultimately cause disease.

3) Discussion Lines 456-457: The authors state that “This occurs probably by sarcomere reorganization under controlled contractility.” The “probably” may be to strong as it is without evidence from this study, nor is it referenced.

Data/Statistical Concerns

4) Results and Figures 2,4: The authors note that they did not observe a correlation between cell area and ablation. Similarly, they describe no clear correlation between distance from nucleus to ablation, except in the presence of an interaction with the number of sarcomeres. The legend of the figures are titled “Correlation of CMs…”. However, Figures 2,4 only includes violin plots. A) Please provide statistical results regarding regression/correlations.

B) For figure 4, was an interaction between distance and number of nuclei included in the test? If not, could the authors include such or describe why it is not needed?

5) Results Lines 276-279: The authors report percent changes without statistics. Please clarify whether the data were significantly reduced by ablations or not.

6) Results Lines 369-371: Please indicate the reported power of this test to provide confidence that the reduced shortening is not a Type II (false positive). statistical error

Minor Concerns

7) The authors note throughout that number and distance to the nucleus differed. However, they do not include a hypothesis or justification for looking at nuclei instead of other cellular compartments (for example the endoplasmic reticulum or mitochondria). Please describe why this and/or the finding that 2 nuclei improve survival is an important factor to study.

8) Results Lines 343-344: The authors describe that alignment within a 20 degree angle was compared. Please include a statement of why this range was chosen.

9) Figure 6 Panels C,D: Please check the comparative text to see if the “pre” and “post” labels have been flipped. (E.g. the analysis was unlikely to be done 2 h PRE ablation).

Reviewer #2: The manuscript by Mueller et al. entitled “Evaluation of laser induced sarcomere micro-damage…” is devoted to evaluation of method to introduce micro-damage in sarcomere. This new laser-based approach allows for the precise ablation of individual elements of the sarcomere, i.e. Z-disc. In their previous article (Mueller at al. 2019) the authors established the femtosecond laser-based system with the sub-micrometer precision cell manipulation. Here, they demonstrate correlation between spatially confined micro-damages and cell survival. The problem of the study is very important: many cardiomyopathies develop sarcomere disarray demonstrating cardiomyocyte losses. The femtosecond laser-based nanosurgery followed by the fluorescence microscopy is a powerful tool to better understand integrity and structure-function of the sarcomeric machinery in healthy and disease conditions.

These evaluation assays might be of interest for the researchers who study cardiomyopathies: this is a good model of the sarcomere disarray. These methods will also help to investigate cytoskeleton structure-function and sarcomere reparation.

I suppose that this interesting methodological research article can be published in PLoS One , just some flaws need to be corrected and some important questions should be addressed prior publication.

Major:

I could not find an answer to the following question that should be addressed:

1. Authors should convince a reader that the ablation of the Z-disk really took place.

Rat newborn cardiomyocytes were transfected with anti-a-actinin fused with turbo-RFP for visualization of Z-disks. The ‘free-electron-mediated bond-breaking’ can also affect structure of the RFP reporter, recognition site of antibody etc. Authors should show phase microscopy photographs in parallel to the fluorescence microscopy images for reader to ensure that the damage really took place in the Z-disk proteins.

2. How cell area was measured? It is not clear from the Methods. For the in vivo cardiomyocyte membrane visualization, I would recommend di-4 ANEPPS membrane voltage -sensitive dye.

3. What was the cause of the cell death after ablation(s)? This important question should be addressed in Discussion. If the main purpose of the paper is evaluation of the method, and cell viability is most important parameter, this question at least should be discussed.

Minor:

Intro:

Line 90: ‘influence of location of the damage’. Influence on what?

Results:

Line 233: add words ‘within 2 hrs after damage’ Do the same for the Figure 1 capture. Why did authors choose 2 hrs interval in those experiments? Why in some experiments authors use 24 hrs instead of 2 hrs?

Figure 1. It might be a good idea to show a correlation plot Viability vs # of Z-disk ablations.

Line256: “A critical factor for CMs survival […] could be the size and the total number of Z-disks…” This is not good hypothesis to test. Hypothesis should be something like “CMs with smaller cell area more susceptible to the damage ”

Line 263: ‘Cardiomyocytes with fewer Z-disc counts…’ Very confusing sentence. Need to re-phrase.

Line 277 and 279: Were these differences significant?

Figure 4, panels C and D: Are these differences significant? How to treat these violin plots w/o statistical significance?

Line 308: Why opposite correlation was observed in CMs with two nuclei? Significant?

Figure 5. Mistype: pattern strengTH

Could authors explain for a reader how to read these pattern strength maps? It is not easy.

Line 317: What is ‘continuous’ contractility? Contractility is a property of CMs: it can’t be continuous. Contraction?

Line 330: It is not clear where. Authors should point out the location where 5 Z-disks were ablated.

Figure 6, panels C and D: note the units (% of cells)

Figure 7, panel D: why only single Z-disk damage reflected in sarcomere shortening? It is not convincing for a reader.

Discussion:

Line 387: Did authors study sarcomeregenesis?

Line 397-398: to make this conclusion authors needed to compare randomly to neighboring Z-disks damage. This comparison had not been made.

Lines 443-447. Forces can’t be directly quantified from these experiments.

6. PLOS authors have the option to publish the peer review history of their article (what does this mean?). If published, this will include your full peer review and any attached files.

Reviewer #1: No

Reviewer #2: No

---

## [Author Response · Author response to Decision Letter 0]

2 Mar 2021

We gratefully appreciate the helpful comments and critique of the reviewers on our paper. We have addressed all points raised by the reviewers in the Revision - Response to Decision file.

---

## [Decision Letter · Decision Letter 1]

25 Mar 2021

PONE-D-21-00038R1

Evaluation of laser induced sarcomere micro-damage: role of damage extent and location in cardiomyocytes

PLOS ONE

Dear Dr. Müller,

Thank you for submitting your manuscript to PLOS ONE. After careful consideration, we feel that it has merit but does not fully meet PLOS ONE’s publication criteria as it currently stands. Therefore, we invite you to submit a revised version of the manuscript that addresses the points raised during the review process.

We look forward to receiving your revised manuscript.

Kind regards,

Xiaolei Xu

Academic Editor

PLOS ONE

Reviewers' comments:

Reviewer's Responses to Questions

**Comments to the Author**

1. If the authors have adequately addressed your comments raised in a previous round of review and you feel that this manuscript is now acceptable for publication, you may indicate that here to bypass the “Comments to the Author” section, enter your conflict of interest statement in the “Confidential to Editor” section, and submit your "Accept" recommendation.

Reviewer #1: (No Response)

Reviewer #2: All comments have been addressed

2. Is the manuscript technically sound, and do the data support the conclusions?

Reviewer #1: Partly

Reviewer #2: Yes

3. Has the statistical analysis been performed appropriately and rigorously? 

Reviewer #1: I Don't Know

Reviewer #2: Yes

4. Have the authors made all data underlying the findings in their manuscript fully available?

Reviewer #1: Yes

Reviewer #2: Yes

5. Is the manuscript presented in an intelligible fashion and written in standard English?

Reviewer #1: Yes

Reviewer #2: Yes

6. Review Comments to the Author

Reviewer #1: The submission PONE-D-21-00038R1 by Muller et al, argues that ablation of multiple z-disks using a femtosecond laser is associated with cell death and damage to contractile function, related to mofibril disarray. While the authors have substantial evidence to suggest that the laser treatment causes z-disc misalignment, this reviewer remains unconvinced that the cell death is due to z-disc ablation and not the laser treatment. Nonetheless, the authors report unique methodology and provide improved context and limitations within the discussion.

Major Comment:

1. Statistics: While this reviewer is grateful that the authors improved their reporting of statistical results, the revised results section highlights several concerns. For example, line 281-282 of the highlighted manuscript states "A non

significant decreased viability from 68% (2 h) to 60% (24 h) was observed (p = 0.99)...". It is unclear why the authors would highlight changed percentages when the differences were not statistically significant. Even for data nearing p~0.15, it would seem that the authors are purposely highlighting a difference that is not there. Given the sample sizes, this reviewer would suggest removing statements that state differences or trends in mean differences when the data are not near p=0.05.

Minor Comments:

2. Statistics: Given that Viability (Figure 1C/D) included two factors (control/treated; number of ablations), why are the authors reporting a one-way ANOVA instead of a multi-factor ANOVA?

3. (Related to Reviewer's original Comment 1) While this reviewer understand the author's interest in further pursing the mechanisms of cell death in future studies (and greatly appreciates the inclusion of the new ROS data and expanded discussion), this reviewer is not convinced that the z-disc removal causes the cell death. This reviewer would be more comfortable with the results if they showed that femtosecond laser treatment was not the cause of cell death. (For example, via distributed treatment, as previously suggested, or by treating a non-sarcomeric structure with the same ablation (either in a muscle cell away from nuclei and striations, which is difficult, or a non-muscle cell type).

Reviewer #2: The manuscript by Mueller et al. entitled “Evaluation of laser induced sarcomere micro-damage…” was significantly improved in my opinion.

A couple of things to fix, though:

i) I still do not like lines 259-261. Authors should mention that this is amount of Z-disks in ROI, not total. I would stick to the parameter ‘cell area’ instead of amount of Z-disks. I think that is most suitable.

ii) I would suggest not to make ANY conclusion on insignificant data (Lines 268-269; also lines 314-315). Please, check throughout whole manuscript.

Otherwise the manuscript looks much better and certainly match the publication criteria in the PLoS One journal.

7. PLOS authors have the option to publish the peer review history of their article (what does this mean?). If published, this will include your full peer review and any attached files.

Reviewer #1: No

Reviewer #2: No

---

## [Author Response · Author response to Decision Letter 1]

26 Mar 2021

We gratefully appreciate the helpful comments and critique of the reviewers on our manuscript. We have addressed all points raised by the reviewers in the Response to Reviewers file.

---

## [Decision Letter · Decision Letter 2]

19 Apr 2021

PONE-D-21-00038R2

Evaluation of laser induced sarcomere micro-damage: role of damage extent and location in cardiomyocytes

PLOS ONE

Dear Dr. Müller,

Thank you for submitting your manuscript to PLOS ONE. After careful consideration, we feel that it has merit but does not fully meet PLOS ONE’s publication criteria as it currently stands. Therefore, we invite you to submit a revised version of the manuscript that addresses the points raised during the review process.

We look forward to receiving your revised manuscript.

Kind regards,

Xiaolei Xu

Academic Editor

PLOS ONE

Journal Requirements:

Reviewers' comments:

Reviewer's Responses to Questions

**Comments to the Author**

1. If the authors have adequately addressed your comments raised in a previous round of review and you feel that this manuscript is now acceptable for publication, you may indicate that here to bypass the “Comments to the Author” section, enter your conflict of interest statement in the “Confidential to Editor” section, and submit your "Accept" recommendation.

Reviewer #1: (No Response)

Reviewer #2: All comments have been addressed

2. Is the manuscript technically sound, and do the data support the conclusions?

Reviewer #1: Yes

Reviewer #2: Yes

3. Has the statistical analysis been performed appropriately and rigorously? 

Reviewer #1: Yes

Reviewer #2: Yes

4. Have the authors made all data underlying the findings in their manuscript fully available?

Reviewer #1: Yes

Reviewer #2: Yes

5. Is the manuscript presented in an intelligible fashion and written in standard English?

Reviewer #1: Yes

Reviewer #2: Yes

6. Review Comments to the Author

Reviewer #1: The authors, Muller et al, further revise PONE-D-21-00038R2, "Evaluation of laser induced sarcomere micro-damage: role of damage extent and location in cardiomyocytes". The authors provide data suggesting that z-disc ablation using a femtosecond laser is associated with both cell death and myofibril re-alignment.

The authors have addressed the most substantial concerns, but this reviewer suggests further edits on statistical reporting for one result:

Line 278/279 (results for figure 3). It does not make sense to this reviewer that a p=0.99 and 0.59 be highlighted as "a non-significant decreased viability". Since the graphs do visually show stronger trends, this reviewer suggest re-evaluating the data used to derive this statistic. If the p-values are replicated, this reviewer suggest that the authors highlight that 'despite a visual difference' there was no statistically significant difference in viability, instead of maintaining the wording in the current draft.

Reviewer #2: I am satisfied how the reviewer's questions were addressed by the authors. I think the manuscript deserved to be published in PLoS One.

7. PLOS authors have the option to publish the peer review history of their article (what does this mean?). If published, this will include your full peer review and any attached files.

Reviewer #1: No

Reviewer #2: No

---

## [Author Response · Author response to Decision Letter 2]

19 Apr 2021

We gratefully appreciate the helpful comments and critique of the reviewers on our manuscript. We have addressed all points raised by the reviewers in the Response to Reviewers file.

---

## [Decision Letter · Decision Letter 3]

14 May 2021

Evaluation of laser induced sarcomere micro-damage: role of damage extent and location in cardiomyocytes

PONE-D-21-00038R3

Dear Dr. Müller,

We’re pleased to inform you that your manuscript has been judged scientifically suitable for publication and will be formally accepted for publication once it meets all outstanding technical requirements.

Kind regards,

Xiaolei Xu

Academic Editor

PLOS ONE

Additional Editor Comments (optional):

Reviewers' comments:

Reviewer's Responses to Questions

**Comments to the Author**

1. If the authors have adequately addressed your comments raised in a previous round of review and you feel that this manuscript is now acceptable for publication, you may indicate that here to bypass the “Comments to the Author” section, enter your conflict of interest statement in the “Confidential to Editor” section, and submit your "Accept" recommendation.

Reviewer #1: All comments have been addressed

2. Is the manuscript technically sound, and do the data support the conclusions?

Reviewer #1: Yes

3. Has the statistical analysis been performed appropriately and rigorously? 

Reviewer #1: Yes

4. Have the authors made all data underlying the findings in their manuscript fully available?

Reviewer #1: Yes

5. Is the manuscript presented in an intelligible fashion and written in standard English?

Reviewer #1: Yes

6. Review Comments to the Author

Reviewer #1: The manuscript PONE-D-21-00038R3 has been further revised and the reviewer thanks the authors for addressing the concerns.

7. PLOS authors have the option to publish the peer review history of their article (what does this mean?). If published, this will include your full peer review and any attached files.

Reviewer #1: No

---

## [Editor Report · Acceptance letter]

19 May 2021

PONE-D-21-00038R3 

Evaluation of laser induced sarcomere micro-damage: role of damage extent and location in cardiomyocytes 

Dear Dr. Müller:

I'm pleased to inform you that your manuscript has been deemed suitable for publication in PLOS ONE. Congratulations! Your manuscript is now with our production department. 

Kind regards, 

on behalf of

Dr. Xiaolei Xu 

Academic Editor

PLOS ONE